# MimicPlay: Long-Horizon Imitation Learning by Watching Human Play

**Chen Wang**[1], **Linxi Fan**[2], **Jiankai Sun**[1], **Ruohan Zhang**[1],
**Li Fei-Fei**[1], **Danfei Xu**[23], **Yuke Zhu**[24†], **Anima Anandkumar**[25†]
[1]Stanford, [2]NVIDIA, [3]Georgia Tech, [4]UT Austin, [5]Caltech, [†]Equal Advising

**Abstract:** Imitation learning from human demonstrations is a promising paradigm for teaching robots manipulation skills in the real world. However, learning complex long-horizon tasks often requires an unattainable amount of demonstrations. To reduce the high data requirement, we resort to *human play data*—video sequences of people freely interacting with the environment using their hands. Even with different morphologies, we hypothesize that human play data contain rich and salient information about physical interactions that can readily facilitate robot policy learning. Motivated by this, we introduce a hierarchical learning framework named MIMICPLAY that learns latent plans from human play data to guide low-level visuomotor control trained on a small number of teleoperated demonstrations. With systematic evaluations of 14 long-horizon manipulation tasks in the real world, we show that MIMICPLAY outperforms state-of-the-art imitation learning methods in task success rate, generalization ability, and robustness to disturbances. Code and videos are available at mimic-play.github.io.

**Keywords:** Imitation Learning, Learning from Human, Long-Horizon Manipulation

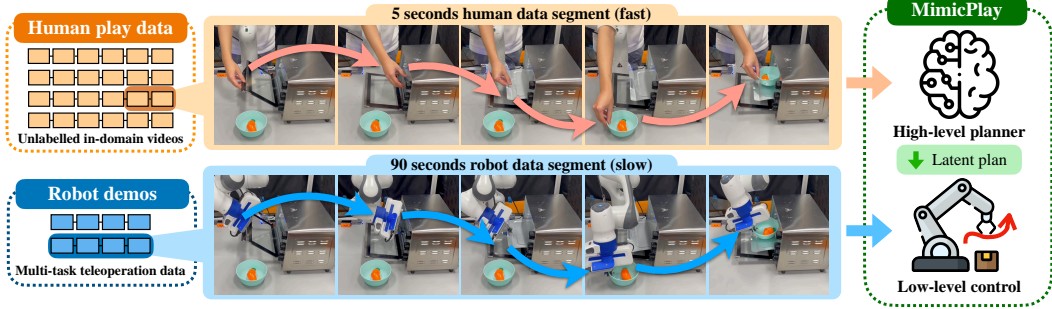

Figure 1: Human is able to complete a long-horizon task much faster than a teleoperated robot. This observation inspires us to develop **MIMICPLAY**, a hierarchical imitation learning algorithm that learns a high-level planner from cheap human play data and a low-level control policy from a small amount of multi-task teleoperated robot demonstrations. We show that MIMICPLAY significantly improves sample efficiency and robustness of imitation learning algorithms in long-horizon tasks.

## 1 Introduction

Efficiently teaching robots to perform general-purpose manipulation tasks is a long-standing challenge. Imitation Learning (IL) has recently made considerable strides towards this goal, especially through supervised training using either human teleoperated demonstrations or trajectories of expert policies [1, 2]. Despite this promise, IL methods have been confined to learning short-horizon primitives, such as opening a door or picking up a specific object. The time cost and labor intensity of collecting long-horizon demonstrations, especially for complex real-world tasks with wide initial and goal condition distributions, remains a key barrier to the widespread adoption of IL methods.

7th Conference on Robot Learning (CoRL 2023), Atlanta, USA.

Two connected directions have emerged in recent literature to scale up imitation learning to complex long-horizon tasks: *hierarchical imitation learning* and *learning from play data*. The former aims to increase learning sample efficiency by decoupling end-to-end deep imitation learning into the learning of high-level planners and low-level visuomotor controllers [3, 4]. The latter leverages an alternative form of robot training data, named *play data* [5]. Play data is typically collected through human-teleoperated robots interacting with their environment without specific task goals or guidance. Prior works show that data collected this way covers more diverse behaviors and situations compared to typical task-oriented demonstrations [5, 6]. The methods for learning from such play data often seek to uncover such diverse behaviors by training a hierarchical policy [5], where the high-level planner captures the distribution of intent and the low-level policies learn goal-directed control. However, collecting such play data in the real world can be very costly. For example, C-BeT [6] requires 4.5 hours of play data to learn manipulation skills in a specific scene, and TACO-RL [7] needs 6 hours of play data for one 3D tabletop environment.

In this work, we argue that the data required for learning high-level plan and low-level control can come in different forms, and doing so could substantially reduce the cost of imitation learning for complex long-horizon tasks. Based on this argument, we introduce a new learning paradigm, in which robots learn high-level plans from *human play data*, where humans use their hands to interact with the environment freely. Human play data is much faster and easier to collect than robot teleoperation data (Fig. 1). It allows us to collect data *at scale* and cover a wide variety of situations and behaviors. We show that such scalability plays a key role in strong policy generalization. The robot then learns low-level manipulation policies from a small amount of *demonstration data*, which is collected by humans teleoperating with the robots. Unlike human play data, demonstration data is expensive to collect but does not lead to issues due to the mismatch between human and robot embodiments.

To scale imitation learning to long-horizon manipulation tasks, we present MIMICPLAY, a new imitation learning algorithm that leverages the complementary strengths of two data sources mentioned above: human play data and robot teleoperation data. MIMICPLAY trains a goal-conditioned latent planner from *human play data* to predict the future 3D human hand trajectories conditioned on the goal images. Such latent plans provide rich 3D guidance (*what* to do and *where* to interact) at each time step, tackling the challenging long-horizon manipulation problem by converting it into a guided motion generation process. Conditioned on these latent plans, the low-level controller incorporates state information essential for fined-grained manipulation to generate the final actions. We evaluate our method on 14 real-world long-horizon manipulation tasks in six environments. Our results demonstrate significant improvement over state-of-the-art imitation learning methods in terms of sample efficiency and generalization abilities. Moreover, MIMICPLAY integrates human motion and robotic skills into a joint latent plan space, which enables an interface that allows using human videos directly as "prompts" for specifying goals in robot manipulation tasks.

To summarize, the main contributions of our work are as follows:

- A **novel paradigm** for learning 3D-aware latent plans from cheap human play data.
- A **hierarchical framework** that trains a plan-guided multi-task robot controller to accomplish challenging long-horizon manipulation tasks sample-efficiently.
- In 14 real-world long-horizon evaluation tasks, MIMICPLAY shows state-of-the-art performance with generalization to novel tasks and robustness against disturbance, which further allows prompting robot motion with human videos.

## 2  Related Work

**Imitation learning from demonstrations.**  Imitation Learning (IL) has enabled robots to successfully perform various manipulation tasks [8–15]. Traditional IL algorithms such as DMP and PrMP [16–19] enjoy high learning sample efficiency but are limited in their ability to handle high-dimensional observations and settings that require closed-loop control. In contrast, recent IL methods built upon deep neural networks can learn reactive policies from raw demonstration data [20, 2, 3, 21–24]. While these methods offer greater flexibility, they require a large number of human demonstrations to learn even simple pick-and-place tasks, which remains labor- and resource-intensive [25, 26]. Our work instead proposes

to leverage human play data, which does not require robot hardware and can be collected efficiently, to reduce the need for on-robot demonstration data dramatically.

**Hierarchical imitation learning.** Our idea of learning a hierarchical policy from demonstrations is also related to prior works [27, 3, 5, 4, 28]. However, all previous methods focus on learning both planning and control with a single type of data—teleoperated robot demonstrations, which is expensive to collect. Our approach uses cheap human play data for learning high-level planning and a small number of robot demonstrations for learning low-level control, which significantly strengthens the model's planning capability while keeping a low demand on demonstration data.

**Learning from human videos.** A plethora of recent research has explored leveraging large-scale human video data to improve robot policy learning [29–39]. Closely related to our work are R3M [40] and MVP [41], which use an Internet-scale human video dataset Ego4D [42] to pretrain visual representations for subsequent imitation learning. However, due to diversity in the data source and large domain gaps, transferring the pre-trained representation to a specific manipulation task might be difficult. Notably, Hansen et al. [43] found simple data augmentation techniques could have similar effects as these pre-trained representations. To reduce the domain gap, another thread of work [29, 44, 35, 45, 34, 46, 47] utilizes in-domain human videos, where human directly interacts with the robot task environment with their own hands. Such type of data has a smaller gap between human and robot domains, which allows sample-efficient reward shaping for training RL agents [35, 45–47] and imitation learning [29, 44, 34]. However, these works focus on learning either task rewards or features from human videos, which doesn't directly help the low-level robot action generation. In this work, we extract meaningful trajectory-level task plans from human play data, which provides high-level guidance for the low-level controller for solving challenging long-horizon manipulation tasks.

**Learning from play data.** Our idea of leveraging human play data is heavily inspired by learning from play [5, 48, 6], an alternative imitation learning paradigm that focuses on multi-task learning from play data, a form of teleoperated robot demonstration provided without a specific goal. Although play data exhibits high diversity in behavior [5], it requires the laborious teleoperation process (4.5 hours [6] and 6 hours [7]). In this work, we instead learn from human play data, where humans freely interact with the scene with their hands. This method of data collection is not only time-effective, requiring a mere 10 minutes, but it also provides rich trajectory-level guidance for the robot's motion generation. Consequently, the robot only requires a minimal amount of teleoperation data, empirically less than 30 minutes, to translate the guidance into its own motor commands and successfully perform complex long-horizon manipulation tasks.

## 3 MimicPlay

Training a robot for long-horizon tasks is challenging, as it requires high-level planning to determine *where* and *what* to interact during different task stages, as well as low-level motor controls to handle *how* to achieve the goals. MIMICPLAY is based on the key insight that high-level planning can be effectively learned from human play data that are fast to collect. Meanwhile, low-level control skills are best acquired from teleoperated demonstration data that do not have any embodiment gap. In particular, due to the difference in the embodiments of humans and robots, it is critical to find an intermediate representation that can bridge the gap between the two data sources. MIMICPLAY addresses this challenge by learning a 3D-aware latent planning space to extract diverse plans from cost-effective human play data. The overview of MIMICPLAY is illustrated in Fig. 2.

### 3.1 Collecting human play data

**Human play data.** We leverage human play data, where a human operator freely interacts with the scene with one hand driven by their curiosity. For instance, in the kitchen, the operator might open the oven then pull out the tray or pick up a pan and place it on the stove. This type of data contains rich state transitions and implicit human knowledge of objects' affordance and part functionalities. More importantly, collecting human play data is cheaper and much more efficient than teleoperation, as it does not require task labeling or environment resetting and takes only a small fraction of time—a human operator can finish a task that would take 90-second robot teleoperation time in just five seconds (Fig. 1). In this work,

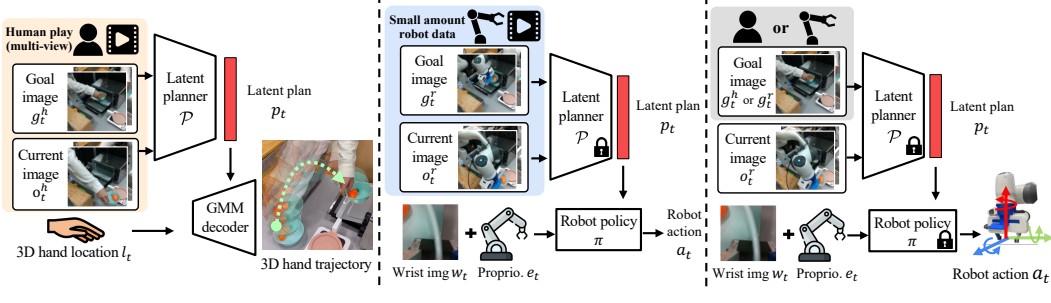

Figure 2: Overview of MIMICPLAY. **(a) Training Stage 1**: using cheap *human play data* to train a goal-conditioned trajectory generation model to build a latent plan space that contains high-level guidance for diverse task goals. **(b) Training Stage 2**: using a small amount of *teleoperation data* to train a low-level robot controller conditioned on the latent plans generated by the pre-trained (frozen) planner. **(c) Testing**: Given a single long-horizon task video prompt (either human motion video or robot teleoperation video), MIMICPLAY generates latent plans and guides the low-level controller to accomplish the task.

we collect 10 minutes of human play video as the training dataset for each task environment, which is approximately equivalent to a 3-hour dataset of robot teleoperation video.

**Tracking 3D human hand trajectories.**    When performing the play, human constructs a sequence of movements within their mind, then engages its hand to physically interact with the environment. This interaction creates a hand trajectory that contains rich information regarding the individual's underlying intentions. Based on the hand trajectories, the robot can learn to mimic human's motion planning capability by reconstructing the hand trajectory conditioned on the goals. We will show how to train a latent planner from human hand trajectory data in Sec. 3.2. However, common human video datasets comprise single-view observations, providing only 2D hand trajectories. Such trajectories present ambiguities along the depth axis and suffer from occlusions. We instead use two calibrated cameras to track 3D hand trajectories from human play data. We use an off-the-shelf hand detector [49] to identify hand locations from two viewpoints, reconstructing a 3D hand trajectory based on the calibrated camera parameters. Details of the data collection process and system are introduced in the Appendix.

## 3.2    Learning 3D-aware latent plans from human play data

Given a long-term task represented by a goal image, the policy should generate actions conditioned on this goal. We formalize the problem into a hierarchical policy learning task, where a goal-conditioned high-level planner $\mathcal{P}$ distills key features from the goal observation $g_t$ and transforms them into low-dimensional latent plans $p_t$. These plans are then employed to guide the low-level motor controller toward the goal. However, learning such a vision-based motion planner $\mathcal{P}$ requires a large dataset since it needs to be capable of handling the multimodality inherent in the goal distribution. We address this issue by leveraging a cheap and easy-to-collect data source—*human play data*.

**Learning multimodal latent plans.**    With the collected human play data and corresponding 3D hand trajectory $\boldsymbol{\tau}$, we formalize the latent plan learning process as a goal-conditioned 3D trajectory generation task. More specifically, an observation encoder $E$, implemented as convolutional networks, processes the visual observations $o_t^h$ and goal image $g_t^h$ from the human video $\mathcal{V}^h$ into low-dimensional features, which are further processed by an MLP-based encoder network into a latent plan vector $p_t$ (as shown in Fig. 2(a)). Based on the latent plan $p_t$ and the hand location $l_t$, an MLP-based decoder network generates the prediction of the 3D hand trajectory. However, simple regression of the trajectory cannot fully cover the rich multimodal distribution of human motions. Even for the same human operator, one task goal can be achieved with different strategies. To address this issue, we use an MLP-based Gaussian Mixture Model (GMM) [50] to model the trajectory distribution from the latent plan $p_t$. For a GMM as Eq. (1) shows:

$$p(\boldsymbol{\tau}|\boldsymbol{\theta}) = \sum_{\boldsymbol{z}} p(\boldsymbol{\tau}|\boldsymbol{\theta}, \boldsymbol{z}) p(\boldsymbol{z}|\boldsymbol{\theta}), \tag{1}$$

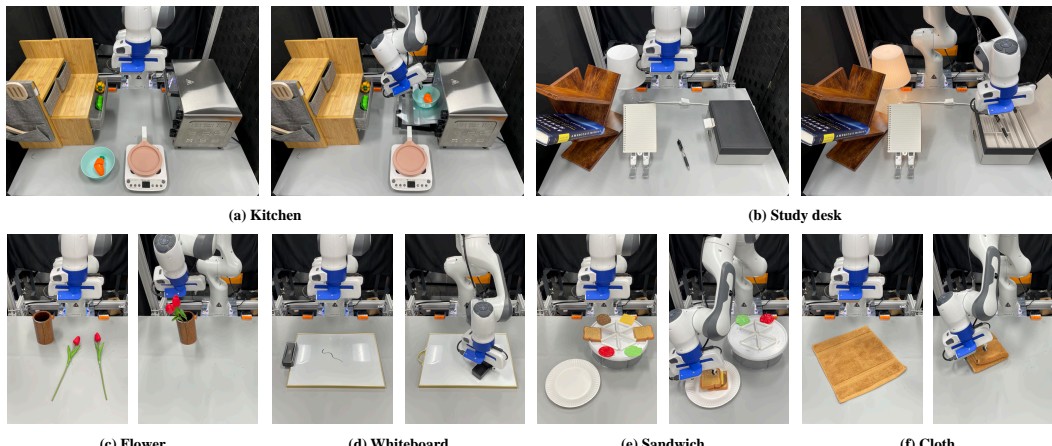

**(a) Kitchen**   **(b) Study desk**

**(c) Flower**   **(d) Whiteboard**   **(e) Sandwich**   **(f) Cloth**

Figure 3: **Evaluation Tasks.** We design six environments with long-horizon tasks for a Franka Emika robot arm, with initial (left) and goal (right) states shown in images. Tasks include: **(a) Kitchen environment**: 3 individual tasks including cooking food with oven. **(b) Study Desk environment**: 7 individual tasks including tidying up the desk. **(c) Flower**: flower insertion into a vase. **(d) Whiteboard**: erasing curve lines. **(e) Sandwich**: ingredient selection for cheeseburger or sandwich. **(f) Cloth**: folding a towel twice. See the Appendix for details.

where $\boldsymbol{\theta} = \{\boldsymbol{\mu}_k, \boldsymbol{\sigma}_k, \eta_k\}_{k=1}^{K}$ are the parameters of the GMM and $p(\boldsymbol{\tau}|\boldsymbol{\theta}, \boldsymbol{z}_k)$ is a Gaussian distribution $\mathcal{N}(\boldsymbol{\tau}|\boldsymbol{\mu}_k, \boldsymbol{\sigma}_k)$ with $\boldsymbol{z}$ consisting of $K$ components. A specific weight $\eta_k$ represents the probability of the $k$-th component. GMMs are more expressive than simple MLPs, because they are designed to capture the multi-modality which is inherited in the human play data. The final learning objective of our GMM model is to minimize the negative log-likelihood of the detected 3D human hand trajectory $\boldsymbol{\tau}$ as Eq. (2)

$$\mathcal{L}_{\text{GMM}}(\boldsymbol{\theta}) = -\mathbb{E}_{\boldsymbol{\tau}}\log\left(\sum_{k=1}^{K}\eta_k\mathcal{N}(\boldsymbol{\tau}|\boldsymbol{\mu}_k, \boldsymbol{\sigma}_k)\right), \text{where } 0 \leq \eta_k \leq 1, \sum_{k=1}^{K}\eta_k = 1 \qquad (2)$$

**Handling visual gap between human and robot domains.**   We consider the setup where the human and the robot interact in the same environment. However, different visual appearances (for example, top vs. bottom row in Fig. 1) between human and robot domains pose a challenge in transferring the learned latent planner to the downstream robot control. We introduce a new learning objective to minimize the visual representation gap between the two domains. Given human video frames $o^h \in \mathcal{V}^h$ and on-robot video frames $o^r \in \mathcal{V}^r$, we calculate the distribution (mean and variance) of the feature embeddings outputted by the visual encoder $E$ of the human domain $\mathcal{Q}^h = E(o^h)$ and the robot domain $\mathcal{Q}^r = E(o^r)$ in each training data batch. We then minimize the distance between $\mathcal{Q}^h$ and $\mathcal{Q}^r$ with a Kullback–Leibler (KL) divergence loss: $\mathcal{L}_{\text{KL}} = D_{\text{KL}}(\mathcal{Q}^r \| \mathcal{Q}^h)$. Note that, our approach does not require paired human-robot video data. $\mathcal{V}^h$ and $\mathcal{V}^r$ can be different behavior and solving different tasks. Only the image frames from the video are used to minimize the representation gap between the two domains. The final loss function for training the latent planner is: $\mathcal{L} = \mathcal{L}_{\text{GMM}} + \lambda \cdot \mathcal{L}_{\text{KL}}$, where $\lambda$ is a hyperparameter that controls the weights between the two losses.

### 3.3   Plan-guided multi-task imitation learning

MIMICPLAY focuses on multi-task imitation learning settings, where a single policy is trained to perform multiple tasks conditioned on different goals. Prior works often learn multi-task visuomotor policies end-to-end from scratch [6, 48, 5]. However, given the large goal space, these methods require a large amount of teleoperation data for policy training (4.5 hours [6] and 6 hours [7]). In this work, we leverage the latent planner $\mathcal{P}$, pretrained with cost-effective human play data (10 minutes), to condense high-dimensional inputs into low-dimensional latent plan vectors $p_t$. Since these latent plans $p_t$ can offer rich 3D guidance for formulating low-level robot actions $a_t$, the low-level policy $\pi$ can focus on learning the conversion between the low-dimensional plans $p_t$ and actions $a_t$ - a task it can learn efficiently due

| | Subgoal (first subgoal) | | | | | | | | Long horizon ($\geq$ 3 subgoals) | | | | | | | |
|---|---|---|---|---|---|---|---|---|---|---|---|---|---|---|---|---|
| | 20 demos | | | | 40 demos | | | | 20 demos | | | | 40 demos | | | |
| | Task-1 | Task-2 | Task-3 | ALL | Task-1 | Task-2 | Task-3 | ALL | Task-1 | Task-2 | Task-3 | ALL | Task-1 | Task-2 | Task-3 | ALL |
| GC-BC (BC-RNN) [20] | 0.1 | 0.0 | 0.1 | 0.07 | 0.1 | 0.2 | 0.2 | 0.17 | 0.0 | 0.0 | 0.0 | 0.00 | 0.0 | 0.0 | 0.1 | 0.03 |
| GC-BC (BC-trans) [52] | 0.2 | 0.0 | 0.0 | 0.07 | 0.3 | 0.7 | 0.6 | 0.53 | 0.0 | 0.0 | 0.0 | 0.00 | 0.0 | 0.0 | 0.1 | 0.03 |
| C-BeT [6] | 0.5 | 0.6 | 0.0 | 0.37 | 0.4 | **1.0** | 0.0 | 0.47 | 0.0 | 0.0 | 0.0 | 0.00 | 0.0 | 0.0 | 0.0 | 0.00 |
| LMP [5] | 0.3 | 0.1 | 0.2 | 0.20 | 0.6 | 0.3 | 0.2 | 0.37 | 0.1 | 0.0 | 0.1 | 0.07 | 0.3 | 0.1 | 0.0 | 0.13 |
| R3M-BC [40] | 0.9 | 0.0 | 0.0 | 0.30 | 0.5 | 0.4 | 0.0 | 0.30 | 0.0 | 0.0 | 0.0 | 0.00 | 0.5 | 0.0 | 0.0 | 0.17 |
| Ours (0% human) | **1.0** | 0.5 | 0.3 | 0.60 | **1.0** | 0.5 | 0.5 | 0.67 | 0.3 | 0.1 | 0.3 | 0.23 | 0.4 | 0.3 | 0.5 | 0.40 |
| Ours | **1.0** | **0.8** | **0.7** | **0.83** | **1.0** | 0.9 | **0.8** | **0.90** | **0.7** | **0.3** | **0.4** | **0.47** | **0.7** | **0.6** | **0.8** | **0.70** |

Table 1: Quantitative evaluation results in the Kitchen environment.

to the decreased dimensionality. In the following, we introduce how to generate the latent plan $p_t$ and the details of training the plan-guided low-level controller $\pi$ with a small amount of data.

**Video prompting for latent plan generation.** Instructing a robot to perform visuomotor long-horizon tasks is challenging due to the complexity of goal specifications. Our latent planner $\mathcal{P}$, learned from human play videos, is capable of interpolating 3D-aware task-level plans directly from human motion videos, which can serve as an interface for promoting long-horizon robot manipulation. More specifically, we use a one-shot video $\mathcal{V}$ (either human video $\mathcal{V}^h$ or robot video $\mathcal{V}^r$) as a goal specification prompt sent to the pre-trained latent planner to generate robot-executable latent plans $p_t$. The one-shot video is first converted into a sequence of image frames. At each time step, the high-level planner $\mathcal{P}$ takes one image from the sequence as a goal-image input $g_t$ and generates a latent plan $p_t$ to guide the generation of low-level robot action $a_t$. After executing $a_t$, the next image frame in the sequence is used as a new goal image. During the training (Fig. 2(a)(b)), the goal image $g_t^r$ ($g_t^r \in \mathcal{V}^r$) is specified as the frame $H$ steps after the current time step in the demonstration. $H$ is a uniformly sampled integer number within the range of $[200,600]$ (10-30 seconds), which performs as a data augmentation process.

**Transformer-based plan-guided imitation.** Decoupling planning from control allows the low-level policy $\pi$ to focus on learning *how* to control the robot by following the guidance $p_t$. The plan-guided imitation learning process is illustrated in Fig. 2(b). However, to execute fine-grained behaviors like grasping an oven handle, merely having high-level guidance is insufficient. It is equally important to consider low-level specifics of the robot end-effector during the action-generation process. Therefore, we convert the robot's wrist camera observation $w_t$ and proprioception data $e_t$ into low-dimensional feature vectors, both with a shape of $\mathbb{R}^{1 \times d}$. We then combine these features with the generated latent plan $p_t$ to create a one-step token embedding $s_t = [w_t, e_t, p_t]$. The sequence of these embeddings over $T$ time steps, $s_{[t:t+T]}$, is processed through a transformer architecture[51] $f_{\text{trans}}$. The transformer-based policy, known for its efficacy in managing long-horizon action generation, produces an embedding of action prediction $x_t$ in an autoregressive manner. The final robot control commands $a_t$ are computed by processing the action feature $x_t$ through a two-layer fully connected network. To address the multimodal distribution of robot actions, we utilize an MLP-based Gaussian Mixture Model (GMM) [50] for action generation. Details regarding the model architecture are outlined in the Appendix.

**Multi-task prompting.** Learning from human play data enables the planner to handle diverse task goals. We demonstrate this empirically by designing all of our evaluation environments to be multi-task and share the same planner $\mathcal{P}$ and the policy $\pi$ models across all tasks in the same environment. For each training sample, the prompting video is uniformly sampled from the training videos of the same task category.

## 4 Experiment Setups

**Environments and Tasks.** We create six environments with 14 tasks (Fig. 3), featuring tasks such as contact-rich tool use, articulated-object handling, and deformable object manipulation. Three tasks are designed for the Kitchen environment and four for the Study Desk environment, focusing on long-horizon tasks with different goals. We assess methods using *Subgoal* and *Long horizon* task categories. The Study Desk environment examines compositional generalization with three unseen tasks: *Easy*, *Medium*, and *Hard*. The *Easy* task combines two trained tasks, while the *Medium* and *Hard* tasks require novel motions for unseen subgoal compositions, i.e., the transition from subgoal $A$ to subgoal $B$ is new. The horizon of all tasks is between 2000 to 4000 action steps, which equals to 100-200 seconds of robot execution (20Hz control frequency). For more details about the environments and simulation results, please refer to the Appendix.

| | Trained tasks | | | | | Unseen tasks | | | |
|---|---|---|---|---|---|---|---|---|---|
| | Task-1 | Task-2 | Task-3 | Task-4 | ALL | Easy | Medium | Hard | ALL |
| GC-BC (BC-trans) [52] | 0.0 | 0.0 | 0.0 | 0.0 | 0.00 | 0.0 | 0.0 | 0.0 | 0.00 |
| LMP [5] | 0.0 | 0.0 | 0.0 | 0.0 | 0.00 | 0.0 | 0.0 | 0.0 | 0.00 |
| Ours (0% human) | 0.2 | 0.3 | 0.1 | 0.2 | 0.20 | 0.2 | 0.1 | 0.0 | 0.10 |
| Ours (50% human) | 0.3 | 0.4 | 0.1 | 0.4 | 0.30 | 0.4 | 0.3 | 0.1 | 0.27 |
| Ours (w/o KL) | 0.3 | **0.7** | 0.3 | 0.2 | 0.38 | 0.4 | 0.2 | 0.0 | 0.20 |
| Ours (w/o GMM) | 0.4 | 0.2 | 0.2 | 0.3 | 0.28 | 0.2 | 0.0 | 0.0 | 0.07 |
| Ours | **0.6** | **0.7** | **0.4** | **0.5** | **0.55** | **0.7** | **0.5** | **0.2** | **0.47** |

Table 2: Ablation evaluation results in the Study Desk environment (20 demos).

| | Spatial generalization | | Extreme long horizon | Deformable | |
|---|---|---|---|---|---|
| | Flower | Whiteboard | Sandwich | Cloth | ALL |
| LMP-single | 0.1 | 0.0 | 0.1 | 0.3 | 0.13 |
| LMP [5] | 0.0 | 0.0 | 0.0 | 0.2 | 0.05 |
| R3M-single | 0.2 | 0.1 | 0.3 | 0.4 | 0.25 |
| R3M [40] | 0.1 | 0.1 | 0.2 | 0.2 | 0.15 |
| Ours-single | **0.5** | **0.5** | 0.6 | 0.7 | **0.58** |
| Ours | 0.4 | 0.2 | **0.8** | **0.8** | 0.55 |

Table 3: Quantitative evaluation results of multi-task learning.

**Baselines.** We evaluate five methods: 1) GC-BC (BC-RNN) and 2) GC-BC (BC-trans), goal-conditioned BC variants of [20] using RNN and GPT-based transformer architectures, respectively; 3) C-BeT [6], an algorithm using Behavior Transformer [53] to learn from teleoperated robot play data; 4) LMP [5], a method that learns to generate plans and actions in an end-to-end fashion from robot play data; and 5) R3M-BC, a goal-conditioned BC variant of [20] using R3M pre-trained visual representation [40]. All methods, including ours, train on the same robot teleoperation demos (20 or 40 demos per task). Besides this common dataset, baselines add an extra 10-minute robot demos, while MIMICPLAY uses 10-minute human play data. The total data collection time is consistent across methods. Task success rate (%) is the primary metric.

## 5 Results

**Learning latent plans from human play data significantly improves performance.** Our method outperforms Ours (0% human) by more than 23% in long-horizon task settings over all trained tasks, as shown in both Tab. 1 (ALL) and 2 (Trained tasks ALL). This result showcases that learning a latent plan space does not need to rely fully on teleoperated robot demonstration data. A 10-minute of cheap and unlabelled human play data brings large improvements in the task success rate and sample efficiency.

**Hierarchical policy is important for learning long-horizon tasks.** Ours (0% human) trained with our two-stage framework outperform prior end-to-end learning methods in the long-horizon task settings by more than 15%, as is shown in Tab. 1 (ALL) and 2 (Trained tasks ALL). This result shows that end-to-end learning for planning and control is less effective than learning to act based on pre-trained latent plans for long-horizon tasks. We also find the same conclusion in simulation results as is introduced in the Appendix.

**Latent plan pre-training benefits multi-task learning.** In Tab. 3, we study how each method performs when training each task with a separate model. For end-to-end learning approaches (e.g., LMP and R3M-BC), training task-specific models will lead to better performance (LMP-single vs. LMP; R3M-single vs. R3M). These results showcase the difficulty of learning multiple tasks with a single model. However, our approach has the smallest performance drop in multi-task training (Ours-single vs. Ours). These findings highlight the advantage of learning plan-guided low-level robot motions based on the pre-trained latent plan space. However, we do observe an uneven performance drop with our method (the success rate of the whiteboard task drops from 0.5 to 0.2). We hypothesize this is due to the reason that the length of the demonstration for the whiteboard task is shorter than the other tasks, which leads to an imbalanced training dataset.

**GMM is crucial for learning latent plans from human play data.** In Tab. 2, our full with GMM model largely outperforms Ours (w/o GMM). Although being trained with full human play data, Ours (w/o GMM) even fails to match the performance of Ours (0% human) in the generalization task settings. We visualize the trajectories generated by all of the model variants in the Appendix, where we found Ours (w/o GMM) has the worst quality of trajectory generation. This result highlights the importance of using the GMM model to handle the multimodal distribution of the hand trajectory when learning the latent plan space from human play data.

**KL loss helps minimize the visual gap between human and robot data.** In Tab. 2, although Ours (w/o KL) baseline outperforms most baselines in trained tasks, its success rate is 17% lower than Ours. In the generalization setting, Ours (w/o

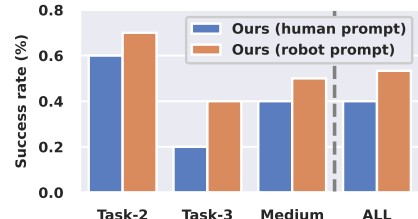

Figure 4: Evaluation of multi-task policy prompted with robot/human videos in the Study Desk environment.

KL) fails to match the performance of Ours (50% human). These results showcase that the visual gap between the human play data and robot data exists, and KL loss helps close the gap when training the latent planner. More analysis of the distribution shifts between human play data and robot data can be found in the Appendix.

**The scale of the human play data matters.**
In Tab. 2, we compared the model variants with 50% human play data (Ours (50% human)) and found it fails to match the performance of Ours, which has access to 100% human play data. Most critically, in the unseen task settings, using more human play data to cover unseen cases in the training set significantly benefits generalization (Ours vs Ours (50% human)).

**Human play data improves generalization to new subgoal compositions.** In Tab. 2 unseen tasks, Ours surpasses all baselines by more than 35%. This result highlights that our approach extracts novel latent plans from human play data and guides the robot's low-level policy to generalize to new compositions of subgoals.

**An intuitive interface for prompting robot motion with human videos.** Fig. 4 shows that our policy model, when prompted with human videos, retains competitive performances as prompted with robot oracle videos across three Study Desk tasks. The reason is that MIMIC-PLAY integrates human motion and robot skills into a joint latent plan space, which enables an intuitive interface for directly specifying robot manipulation goals with human videos. More results can be found in the supplementary video.

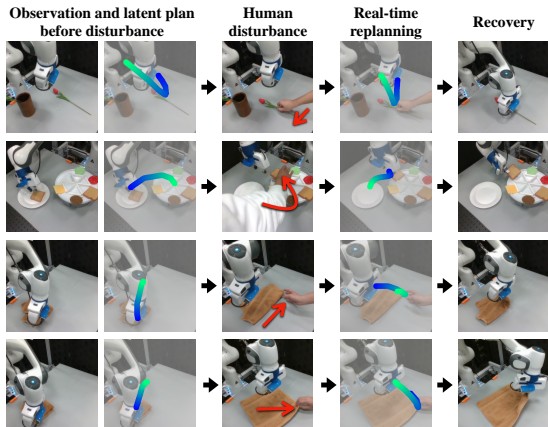

Figure 5: Qualitative visualization of the latent plans before the disturbance and re-planning. **Column 1**: third-person view. **Column 2**: visualization of the latent plan before disturbance. **Column 3**: human disturbance; the red arrow indicts the direction of disturbance. **Column 4**: visualization of real-time re-planning capabilities, which show robustness against disturbance. **Column 5**: robot recovers with the updated task plan.

**Real-time planning capability is robust against disturbance.** In Fig. 5, we showcase how our model reacts to unexpected human disturbance. None of these disturbances appears in the teleoperated robot demonstration data. For instance, in the cloth folding task (Fig. 5 third row), the human unfolds the towel after the robot has folded it, and the robot replans and folds the towel again. Since our whole system (including the vision-based latent planner, low-level guided policy, and robot control) is running at a speed of 17Hz, our model is able to achieve real-time re-planning capability against disturbance and manipulation errors. For more details, please refer to the supplementary video.

## 6 Conclusion and Limitations

Existing limitations of the MIMICPLAY include: 1) The current high-level latent plan is learned from scene-specific human play data. The scalability of MIMICPLAY can greatly benefit from training on Internet-scale data. 2) The current tasks are limited to table-top settings. However, humans are mobile and their navigation behaviors contain rich high-level planning information. The current work can be extended to more challenging mobile manipulation tasks, and 3) There is plenty of room to improve on the cross-embodiment representation learning. Potential future directions include temporal contrastive learning [54] and cycle consistency learning [55] from videos.

We introduce MIMICPLAY, a scalable imitation learning algorithm that exploits the complementary strengths of two data sources: cost-effective human play data and small-scale teleoperated robot demonstration. Using human play data, the high-level controller learns goal-conditioned latent plans by predicting future 3D human hand trajectories given the goal image. Using robot demonstration data, the low-level controller then generates the robot actions from the latent plans. With this hierarchical design, MIMICPLAY outperforms prior arts by over 50% in 14 challenging long-horizon manipulation tasks. MIMICPLAY paves the path for future research to scale up robot imitation learning with affordable human costs.

## Acknowledgments

This work is partially supported by ONR MURI N00014-21-1-2801. L. F-F is partially supported by the Stanford HAI Hoffman-Yee Research Grant. This work is done during Chen Wang's internship at NVIDIA. Stanford provides the computing resources and robot hardware for this project. We are extremely grateful to Yifeng Zhu, Ajay Mandlekar for their efforts in developing the robot control library Deoxys[22] and RoboTurk[56]. We would also like to thank Yucheng Jiang for assisting with multi-seed evaluation in simulation.

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

# A  Implementation details

Here we lay down the details of the data collection, training, and testing process.

**Collecting human play data and training details.** The human play data is collected by letting a human operator directly interact with the scene with a single hand for 10 minutes for each scene. The entire trajectory $\tau$ is recorded at the speed of 60 frames per second and is used without cutting or labeling. The 3D hand trajectory is detected with an off-the-shelf multi-view human hand tracker [49]. The total number of video frames within 10 minutes of human play video is around 36k. We train one latent planner for each environment with the collected human play data. For the multi-environment setup (for the experiments in Tab. 3), we merge the human play data from each scene to train a single latent planner. The latent planner contains two ResNet-18 [57] networks for image processing and MLP-based encoder-decoder networks together with a GMM model, which has $K = 5$ distribution components. We train 100k iterations for the latent planner which takes a single GPU machine for 12 hours.

**Collecting robot demonstrations and training details.** The robot teleoperation data is collected with an IMU-based phone teleoperation system RoboTurk [56]. The control frequency of the robot arm is 17-20Hz and the gripper is controlled at 2Hz. For each task, we collect 20 demonstrations. In the experiments, we also have a 40 demonstration dataset for testing the sample efficiency of different approaches. The robot policy model is a GPT-style transformer [52], which consists of four multi-head layers with four heads. We train 100k iterations for the policy with a single GPU machine in 12 hours. For a fair comparison with our method, the baseline approaches trained without human play data have five more demonstrations during training the latent planner $\mathcal{P}$ and the low-level policy $\pi$.

**Video prompting.** In this work, we use a one-shot video $\mathcal{V}$ (either human video $\mathcal{V}^h$ or robot video $\mathcal{V}^r$) to prompt the pre-trained latent planner to generate corresponding plans $p_t = \mathcal{P}(o_t, g_t, l_t), g_t \in \mathcal{V}$. During training (Fig. 2(b)), we specify the goal image $g_t^r$ ($g_t^r \in \mathcal{V}^r$) as the frame $H$ steps after the input observation $o_t^r$ in the robot demonstration. $H$ is a uniformly sampled integer number within the range of $[200, 600]$, which equals 10-30 seconds in wall-clock time. $l_t$ here is the 3D location of the robot's end-effector. During inference (Fig. 2(c)), we assume access to a task video (either human or robot video) which is used as a source of goal images. The goal image will start at the 200 frame of the task video and move to the next $i$ frame after each step. We use $i = 1$ in all our experiments. Based on the inputs, the latent planner generates a latent plan feature embedding $p_t$ of shape $\mathbb{R}^{1 \times d}$, which is used as guidance for the low-level robot policy.

**Data visualization.** We visualize the collected human play data and robot demonstration data in Tab. 9. For the human play data, we use an off-the-shelf hand detector [49] to localize the hand's 2D location on the left and right image frame, which are visualized as red bounding boxes in Tab 9. For the robot demonstration data, we directly project the 3D location of the robot end-effector to the left and right image frames, which are visualized as blue bounding boxes in Tab 9.

**Testing.** We perform real-time inference on a Franka Emika robot arm with a control frequency of 17Hz—directly from raw image inputs to 6-DoF robot end-effector and gripper control commands with our trained models. The robot is controlled with the Operational Space Control (OSC) [58].

# B  Experiment setups

**Environments.** We design six environments with a total of 14 tasks for a Franka Emika robot arm, as illustrated in Fig. 3. These environments feature several manipulation challenges, such as contact-rich tool manipulation (cleaning the whiteboard), articulated-object manipulation (opening the oven and the box on the study desk), high-precision tasks (inserting flowers and turning on the lamp by pressing the button), and deformable object manipulation (folding cloth).

**Tasks.** We design three tasks in the Kitchen environment and four tasks in the Study desk environment. All these tasks have different goals. In this work, we focus on long-horizon tasks that require the robot to complete several subgoals. To better analyze the performance of each method, we define the *Subgoal* task category that only counts whether the first subgoal of the task has been achieved and the *Long horizon* task category which is the full task. In the Study desk environment, we design three tasks for testing the

| | Subgoal (first subgoal) | | | | | Long Horizon ($\geq$ 3 subgoals) | | | | |
|---|---|---|---|---|---|---|---|---|---|---|
| | Task-1 | Task-2 | Task-3 | Task-4 | Task-5 | Task-1 | Task-2 | Task-3 | Task-4 | Task-5 |
| GC-BC (BC-RNN) | **0.96**±0.02 | 0.00±0.00 | 0.00±0.00 | 0.00±0.00 | 0.00±0.00 | 0.00±0.00 | 0.00±0.00 | 0.00±0.00 | 0.00±0.00 | 0.00±0.00 |
| GC-BC (BC-trans) | 0.95±0.03 | 0.01±0.02 | 0.00±0.00 | 0.01±0.02 | 0.00±0.00 | 0.00±0.00 | 0.00±0.00 | 0.00±0.00 | 0.00±0.00 | 0.00±0.00 |
| Ours (0% human) | 0.92±0.06 | **0.70**±0.05 | **0.80**±0.07 | **0.74**±0.11 | **0.78**±0.04 | **0.58**±0.04 | **0.29**±0.10 | **0.62**±0.11 | **0.59**±0.05 | **0.67**±0.09 |

Table 4: Quantitative evaluation results in simulation (success rates % averaged over 5 seeds)

compositional generalization ability of the models to novel task goal sequences, which are not included in the training dataset. These three tasks are classified as *Easy*, *Medium*, and *Hard* depending on their difference compared to the training tasks. The *Easy* task is a simple concatenation of two trained tasks and their subgoals. The *Medium* task contains an unseen composition of a pair of subgoals that is not covered by any trained tasks, i.e., the transition from subgoal $A$ to subgoal $B$ is new. The model needs to generate novel motions to reach these subgoals. The *Hard* task contains two such unseen transitions. For the rest four environment, each scene has one task goal and features different types of challenges in manipulation, *e.g.*, generalization to new spatial configuration, extremely long horizon, and deformable object manipulation.

**Baselines.** We compare with five prior approaches: (1). GC-BC (BC-RNN) [20]: Goal-conditioned behavior cloning algorithm [5] implemented with recurrent neural networks (RNN) [59]. (2). GC-BC (BC-trans) [52]: Another goal-conditioned behavior cloning algorithm implemented with GPT-like transformer architecture. (3). C-BeT [6]: Goal-conditioned learning from teleoperated robot play data algorithm implemented with Behavior Transformer (BeT) [53]. (4). LMP [5]: A learning from teleoperated robot play data algorithm designed to handle variability in the play data by learning an embedding space. LMP (single) is a variant by training each task with a separate model. (5). R3M-BC [40]: A goal-conditioned imitation learning framework that leverages R3M visual representation pre-trained with internet-scale human video dataset Ego4D [42]. R3M-BC (single) is a variant by training each task with a separate model.

**Ablations.** We compare four variants of our model to showcase the effectiveness of our architecture design: (1). Ours: MIMICPLAY with full collection (10 min) of human play data. Ours (single) is a variant by training each task with a separate model. (2). Ours (0% human): variant of our model without using human play data. The pre-trained latent plan space is trained only with the teleoperated robot demonstrations. (3). Ours (50% human): variant of our model where the latent planner is trained with 50% of human play data (5 min). (4). Ours (w/o GMM): variant without using the GMM model for learning the latent plan space from human play data. (5). Ours (w/o KL): Our approach without using KL loss for addressing the visual gap between human and robot data when pre-training the latent planner.

## C   Supplementary Experiment Results

**Results in simulation.** To extensively evaluate the methods with more testing trials and training seeds, we conduct an experiment in simulation LIBERO [60], which is a multitask robot manipulation benchmark based on robosuite [61] and MuJoCo [62]. We choose LIBERO due to its utilization of the BDDL language [63] for goal specification, which facilitates the multitask evaluation for learning from play data. Note that, in our main paper, we leverage human play data. However, in simulation, there is no way to get such dataset, which will always end up being robot teleoperation. Therefore, in this experiment, we use the same teleoperated robot play dataset to train both high-level planner and low-level controller, and report the results of Ours (0% human) and baselines in Table 4. For each method, we train with 5 random seeds and report the average success rate over 100 testing trials. The results showcase the advantage of MIMICPLAY's hierarchical policy learning framework over the baselines, which is consistent with the real-world results (Tab. 1, 2). The implementation code is available at https://github.com/j96w/MimicPlay.

**Visualization of the trajectory prediction results.** We visualize the 3D trajectory decoded from the latent plan by projecting it onto the 2D image in Fig. 6. In the last two rows, we showcase the results of

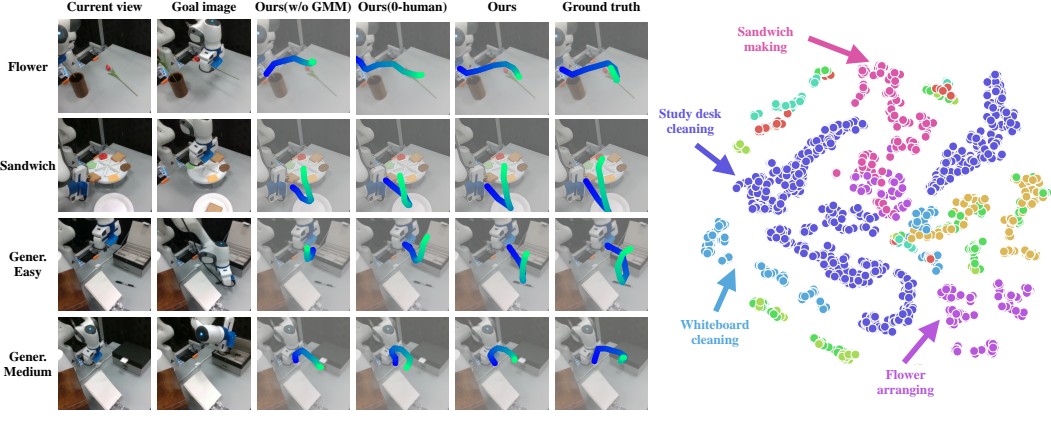

(a) Trajectory prediction results decoded from the latent plans      (b) t-SNE visualization of the latent plans

Figure 6: Qualitative visualization of the learned latent plan. (**a**) Visualization of the trajectory prediction results decoded from the latent plans learned by different methods. The fading color of the trajectory from blue to green indicates the time step from 1 to 10. (**b**) t-SNE visualization of latent plans, the latent plans of the same task tend to cluster in the latent space.

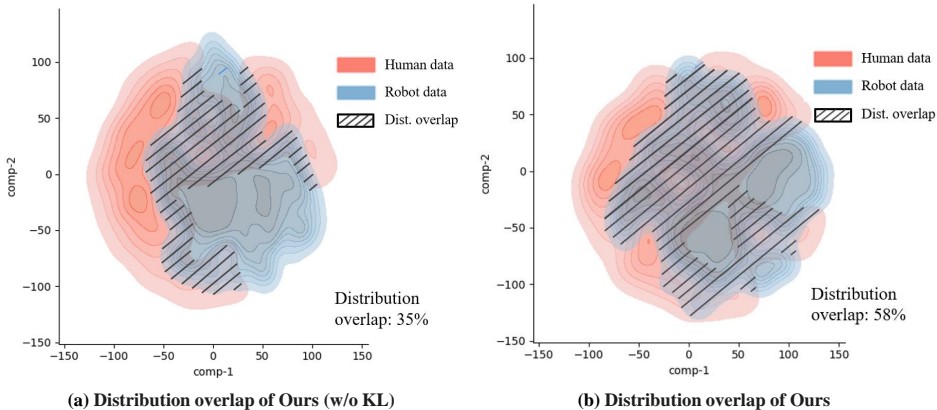

(a) Distribution overlap of Ours (w/o KL)          (b) Distribution overlap of Ours

Figure 7: t-SNE visualization of the generated feature embeddings by taking human data and robot data as inputs. The slashes refer to the overlap region of two data distributions. (**a**) Feature visualization results of our method without using KL divergence loss. (**b**) Feature visualization results of our method with KL divergence loss. Our approach covers 23% more area than the baseline.

two unseen subgoal transitions. The trajectory generated by our model is most similar to the ground truth trajectory, while Ours (0% human) is overfitted to the subgoal transitions in the training set and generates the wrong latent plan. For instance, in the training data, the robot only learns to open the box after turning off the lamp, meanwhile in the *Easy* setting of generalization tasks, the robot is prompted to pick up the pen after turning off the lamp. Ours (0% human) variant still outputs a latent plan to open the box, which causes the task to fail since the box is already open.

**Transformer architecture helps multi-task learning.** In Tab. 1, GC-BC (BC-trans) with the GPT transformer architecture outperforms GC-BC (BC-RNN) by more than 30% in a 40-demos Subgoal setting. However, the performance of GC-BC (BC-trans) quickly drops to the same level as GC-BC (BC-RNN) in

20-demos settings. The result showcases that training vision-based transformer policy end-to-end requires more data.

**Visualization of the learned latent plans.** We use t-SNE [64] to visualize the generated latent plans conditioned on different tasks, as shown in Fig. 6(b). We find that the latent plans of the same task tend to cluster in the latent space, which shows the effectiveness of our approach in distinguishing different tasks.

**Analysis of the visual gap between human and robot data.** As is introduced in the method Sec. 3.2, to minimize the visual gap between human play data and robot demonstration data, we use a KL divergence loss over the feature embeddings outputed by the visual encoders. In Fig. 7, we use t-SNE to process and visualize the learned feature embeddings generated by Ours and the model variant Ours (w/o KL) on the 2D distribution plots. To better visualize the distribution overlap, we use slashes to highlight the overlap area in both plots. We observe that our approach with KL loss has a 23% larger overlap between the human data and the robot data compared to Ours (w/o KL). This result showcases the effectiveness of our KL divergence loss and supports the result in Tab. 2 (Ours (w/o KL) is inferior to Ours in task success rate).

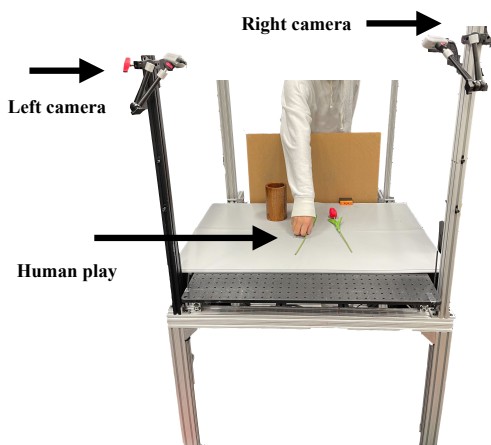

**(a) Human play data collection**

# D   Details of system setups

We illustrate the system designs for the data collection in Fig. 8. The human play data is collected by having a human operator directly interact with the environment with one of its hands (Fig. 8(a)). The left and right cameras record the video at the speed of 100 frames per second. During the collection process of human play data, no specific task goal is given and the human operator freely interacts with the scene for interesting behaviors based on its curiosity. For each scene in our experiments, we collect 10 minutes of human play data.

The robot teleportation demonstration is collected with a phone teleoperation system Robo-Turk [56] (Fig. 8(b)). The left, right, and end-effector wrist cameras record the video at the speed of 20 frames per second, which is aligned with the control speed of the robot arm (20Hz).

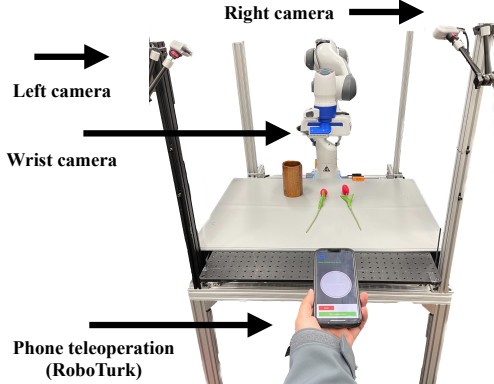

**(b) Robot demonstration data collection**

Figure 8: System setups for the data collection. (**a**) Human play data collection. A human operator directly interacts with the scene with one of its hand and perform interesting behaviors based on its curiosity without a specific task goal. (**b**) Robot demonstration data collection. A human demonstrator uses a phone teleoperation system to control the 6 DoF robot end-effector. The gripper of the robot is controlled by pressing a button on the phone interface.

Each sequence of robot demonstration has a pre-defined task goal. During the data collection, the human demonstrator completes the assigned sub-goals one by one and finally solves the whole task. For each training task in our experiments, we collect 20 demonstrations. In the Kitchen environment, we collect 40 demonstrations for each task to figure out which approach is more sample inefficiency.

# E Details of the task designs

The definition of our long-horizon tasks is listed below. For each task, the initial state and subgoals are pre-defined. The whole task is completed if and only if all subgoals are completed in the correct order.

## E.1 Kitchen

- *Task-1*
    - Initial state: A drawer is placed on the left side of the table. The drawer is not fully open and contains pumpkin and lettuce. A closed microwave oven is placed on the right side of the desktop. A bowl and a stove are placed on the lower edge of the tabletop. There is a carrot inside the bowl. A pan is placed on top of the stove.
    - Subgoals: a) Open the microwave oven door. b) Pull out the microwave oven tray. c) Pick up the bowl. d) Place the bowl on the microwave tray.

- *Task-2*
    - Initial state: same as Kitchen Task-1.
    - Subgoals: a) Open the drawer. b) Pick up the carrot. c) Put the carrots in the drawer.

- *Task-3*
    - Initial state: same as Kitchen Task-1.
    - Subgoals: a) Pick up the pan. b) Place the pan on the table. c) Pick up the bowl. d) Place the bowl on the stove.

## E.2 Study desk

- *Task-1*
    - Initial state: The book is on the rack. The lamp is on. The box is opened and closed in a random state. The pen is located either in the center of the table or in the box.
    - Subgoals: a) Turn off the lamps. b) Pick up the book. c) Place the book on the shelf position.

- *Task-2*
    - Initial state: The location of the book is either on the shelf or on the rack. The lamp is off. The box is closed. The pen is in the center of the table.
    - Subgoals: a) Turn on the lamps. b) Open the box. c) Pick up the pen. d) Put it in the box.

- *Task-3*
    - Initial state: The book is on the rack. The state of the lamp is random. The box is closed. The pen is in the center of the table.
    - Subgoal a) Open the box. b) Pick up the pen. c) Place the pen in the box. d) Pick up the book. e) Place the book on the shelf.

- *Task-4*
    - Initial state: The location of the book is either on the shelf or on the rack. The lamp is on. The box is closed. The pen is located either in the center of the table or in the box.
    - Subgoals: a) Open the box. b) Turn off the lamp.

- *Easy*
    - Initial state: The location of the book is either on the shelf or on the rack. The lamp is off. The box is closed. The pen is located either in the center of the table or in the box.
    - Subgoals: a) Turn on the lamp. b) Open the box. c) Turn off the lamp.

- *Medium*
    - Initial state: The location of the book is either on the shelf or on the rack. The lamp is on. The box is closed. The pen is in the center of the table.

– Subgoals: a) Open the box. b) Turn off the lamp. c) Pick up the pen. d) Place the pen in the box.

- *Hard*

    – Initial state: The book is on the shelf. The lamp is on. The box is closed. The pen is located either in the center of the table or in the box.

    – Subgoals: a) Turn off the lamp. b) Open the box. c) Pick up the book. d) Place the book on the shelf.

### E.3   Flower

- Initial state: Two flowers and a vase are placed on the table. The vase will randomly be placed on the top left or top right corner of the table.

- Subgoals: a) Picking up a flower. b) Insert the flower into the vase. c) Pick up the other flower. d) Insert the flower into the vase.

### E.4   Whiteboard

- Initial state: A whiteboard and board eraser are placed on the table. The board eraser is placed on the left side of the whiteboard.

- Subgoals: a) Pick up the board eraser. b) Moves over the curve line. c) Erase the curve line. d) Return the eraser to the original location.

### E.5   Sandwich

- Initial state: A circular ingredient selector is placed in the upper right corner of the table. Half of the circle holds ingredients for a sandwich (bread, lettuce, sliced tomato) and half holds ingredients for a cheeseburger (bread, cheese, burger patty). A white plate is placed in the lower left corner of the table.

- Subgoals for a sandwich: a) Rotate the ingredient selector to the right position. Pick up a piece of bread from it and place it on the plate. b) Rotate the ingredient selector to the correct position. Pick up the lettuce and place it on top of the bread. c) Rotate the ingredient selector to the right position. Pick up the sliced tomato and place it on top of the lettuce. d) Rotate the ingredient selector to the right position. Pick up another piece of bread and place it on top of the tomato.

### E.6   Cloth

- Initial state: An unfolded brown cloth is randomly placed on the table.

- Subgoals: a) The robot folds the cloth in half once to become 1/2 of its original size. b) The robot folds the cloth once more to become 1/4 of its original size.

## F   Training hyperparameters

We list the hyperparameters for training the models in Tab. 5 for the latent planner $\mathcal{P}$ and Tab. 6 for the robot policy $\pi$. The hyperparameters that are named starting with GMM are related to the MLP-based GMM model. The hyperparameters that are named starting with GPT are related to the transformer architecture. We also list the hyperparameters for the baseline GC-BC (BC-trans) in Tab. 7.

| Hyperparameter | Default |
| --- | --- |
| Batch Size | 16 |
| Learning Rate (LR) | 1e-4 |
| Num Epoch | 1000 |
| LR Decay | None |
| KL Weights $\lambda$ | 1000 |
| MLP Dims | [400, 400] |
| Image Encoder - Left View | ResNet-18 |
| Image Encoder - Right View | ResNet-18 |
| Image Feature Dim | 64 |
| GMM Num Modes | 5 |
| GMM Min Std | 0.0001 |
| GMM Std Activation | Softplus |

Table 5: Hyperparameters - Ours (Latent Planner $\mathcal{P}$)

| Hyperparameter | Default |
| --- | --- |
| Batch Size | 16 |
| Learning rate (LR) | 1e-4 |
| Num Epoch | 1000 |
| Train Seq Length | 10 |
| LR Decay Factor | 0.1 |
| LR Decay Epoch | [300, 600] |
| MLP Dims | [400, 400] |
| Image Encoder - Wrist View | ResNet-18 |
| Image Feature Dim | 64 |
| GMM Num Modes | 5 |
| GMM Min Std | 0.01 |
| GMM Std Activation | Softplus |
| GPT Block Size | 10 |
| GPT Num Head | 4 |
| GPT Num Layer | 4 |
| GPT Embed Size | 656 |
| GPT Dropout Rate | 0.1 |
| GPT MLP Dims | [656, 128] |

Table 6: Hyperparameters - Ours (Robot Policy $\pi$)

| Hyperparameter | Default |
| --- | --- |
| Batch Size | 16 |
| Learning rate (LR) | 1e-4 |
| Num Epoch | 1000 |
| Train Seq Length | 10 |
| LR Decay Factor | 0.1 |
| LR Decay Epoch | [300, 600] |
| MLP Dims | [400, 400] |
| Image Encoder - Wrist View | ResNet-18 |
| Image Encoder - Left View | ResNet-18 |
| Image Encoder - Right View | ResNet-18 |
| Image Feature Dim | 64 |
| GMM Num Modes | 5 |
| GMM Min Std | 0.01 |
| GMM Std Activation | Softplus |
| GPT Block Size | 10 |
| GPT Num Head | 4 |
| GPT Num Layer | 4 |
| GPT Embed Size | 656 |
| GPT Dropout Rate | 0.1 |
| GPT MLP Dims | [656, 128] |

Table 7: Hyperparameters - GC-BC (BC-trans)

# G    Network Architecture

**Transformer-based policy network.** The embedding sequence of $T$ time steps is represented as $s_{[t:t+T]} = [w_t, e_t, p_t, \cdots, w_{t+T}, e_{t+T}, p_{t+T}]$, which passes through a transformer architecture [51]. The transformer model $f_{\text{trans}}$ processes the input embeddings using its $N$ layers of self-attention and feed-forward neural networks. Given an embedding sequence of $T-1$ time steps, $f_{\text{trans}}$ generates the embedding of trajectory prediction in an autoregressive way - $x_T = f_{\text{trans}}(w_{1:T-1}, e_{1:T-1}, p_{1:T-1})$, where $x_T$ is the predicted action embedding at time step $T$. The transformer architecture uses the multi-head self-attention mechanism to gather context and dependencies from the entire history trajectory at each step. The final robot control commands $a_t$ are computed by processing the action feature $x_t$ with a two-layer fully-connected network. To handle the multimodal distribution of robot actions, we also use a MLP-based GMM model [50] for the action generation.

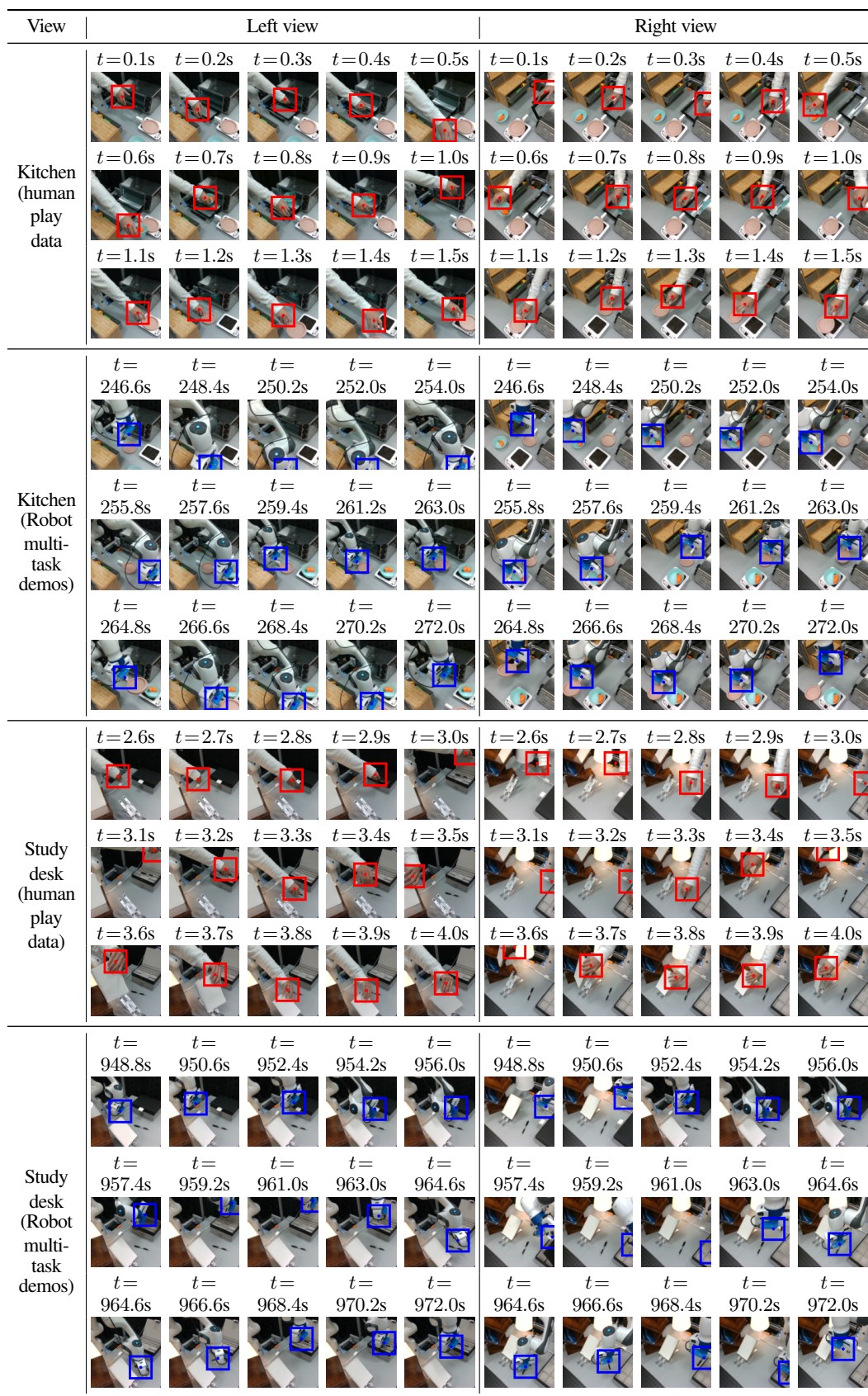

Figure 9: Dataset visualization.

