# OpenReview forum: "MimicPlay: Long-Horizon Imitation Learning by Watching Human Play"
_robot-learning.org/CoRL/2023/Conference — CoRL 2023 Oral_

### Official Review · Reviewer_5FXC · 2023-06-26

**Confidence:** 5
**Originality:** Excellent
**Technical Quality:** Excellent
**Clarity Of Presentation:** Excellent
**Impact:** 4

**Recommendation:**

Strong Accept: I recommend accepting the paper and will argue for my recommendation even if other reviewers hold a different opinion.

**Review:**

## Strengths
* Well motivated novel approach. Unlabeled in-domain human play data indeed seems like a promising direction for future scaling, and this work proposes a clever way of incorporating this type of data into a hierarchical imitation framework. In particular, the use of 3d tracked hand trajectories as high-level pretraining supervision seems well motivated.
* Evidence that the proposed method can indeed incorporate human play data to obtain higher task performance in real world manipulation scenarios, and that more human play data leads to better final policies.
* Solid, well designed experiments and ablations, e.g. studying the effect of human data scale or the impact of the KL term in solving for correspondence mismatch.
* Evidence that the proposed approach can be conditioned with either robot goal images or human goal images, possibly allowing for more flexible test time conditioning scenarios.
* Clear description of methodology and setting.
* Compelling real world results.

## Weaknesses
* Strong claims of significance without evidence of statistical significance. The paper makes multiple claims of “significant improvement” (e.g. L54, L227) without presenting confidence intervals run over multiple trials. While it is perhaps understandable that multiple trials were not run for each real world experiment since it is expensive to do so, claims of significance should be toned down. For example, it might be helpful is the variance in success of the same model run twice in the real world on the same suite of benchmarks? 1%? 10%? 20%?

## Nit
* L121: “we collect 10 minutes of human play video as the training dataset for each task environment, which is approximately equivalent to a 3-hour dataset of robot teleoperation video.” What is the conversion rate here? It would be helpful to describe how 10 minutes of human play is equivalent to 3 hours of robot teleoperation in more detail.

**Quality Of The Limitations Section:**

Limitations are addressed clearly

**Questions For Rebuttal:**

The main question I have is: are any of the real world results leveraging human data statistically significantly better than baseline methods, but it is understandable if these answers are difficult to come by in real world evals.

**Robotics Focus:**

Sufficient demonstration on hardware

**Summary Of Paper:**

* Allows robots to leverage less burdensome unlabeled in-domain human play data to learn high level plans, while learning low-level plan execution from labeled multitask robot demonstrations.
* Same method allows for conditioning directly on human video to guide task execution.


**Summary Of Recommendation:**

This work presents compelling real world evidence that large amounts of in-domain human play data can improve the performance of a multitask imitation learning robot doing difficult long-horizon manipulation tasks. It feels important to broadcast this conclusion to the community, as it may be an important and scalable future resource for robot learning, and may open the door to follow-on work such as learning from out of domain human play data.

---

### Official Review · Reviewer_4uup · 2023-07-16

**Confidence:** 3
**Originality:** Good
**Technical Quality:** Very Good
**Clarity Of Presentation:** Excellent
**Impact:** 4

**Recommendation:**

Strong Accept: I recommend accepting the paper and will argue for my recommendation even if other reviewers hold a different opinion.

**Review:**

Strengths:
* The paper is well written, clearly positioning itself in the landscape of prior work, and clearly explaining the methodology.
* The multimodal high-level latent planner that was decoupled from the low-level control was an intuitive way to model the problem
* Thorough analysis on the many components involved in this work, including robustness against unexpected disturbances and many robotic tasks
* The hardware results and videos are neat!

Weaknesses:
* One of the main weaknesses is that assumptions underlying this approach could be clearer. For example, it is unclear if the multi-view camera assumption in stage 1 is consistent throughout the rest of the stage 1 and testing. It is also unclear if the robot’s wrist camera is used only during stage 2, or in testing. Similarly, it is tough to understand how the “small batch of on-robot data” that is paired with the human play data during stage 1 training is collected. I found these assumptions to be hidden or omitted from the text, but they are central to understanding the impact of this framework.
* Nitpick – there is not an explicit limitations section (it is just part of the conclusion), and the submission instructions requested “All Submissions should include a Limitations section.”

Overall comments:
* “...10 minutes of human play video as the training dataset for each task environment, which is
approximately equivalent to a 3-hour dataset of robot teleoperation video.” How did you arrive at this conversion? Did you empirically observe this?
* “...to address the multimodal distribution of robot actions, we utilize an MLP-based Gaussian Mixture Model (GMM) [50] for action generation” at deployment time, do you sample from this GMM to get the action to execute, or take the MLE? It would be helpful to state how you use the properties of the GMM at deployment time.
* “...H is a uniformly sampled integer number within the range of [200,600], which equals 10-30 seconds in wall-clock time.” How long was the typical task horizon? It would be helpful to know what fraction of the task these subgoals constituted.
* One section that was unclear was how the latent planner – which according to figure 2 is only trained from human play data – is expected to generalize to visual inputs of the robot doing the task. The text seems to imply that “given human video frames and frames from a small batch of on-robot data '' that there is some amount of robot teleoperation data that is used in stage 1 training. However, this seems tough to get in practice. For example, if the human is just executing their natural play data, then to get on-robot data means that another human teleoperator needs to reproduce all the play motions on the robot? This does not seem very scalable or easy to reproduce; at he same time, it seems like a key step to making the planner useful in stage 2, so more details would be appreciated.


**Quality Of The Limitations Section:**

Limitations are addressed clearly

**Questions For Rebuttal:**

* In the video prompting, how sensitive is the method to the H parameter when selecting the goal image? It seems like choosing a good time horizon and the right “image waypoint” would matter a lot for generating a useful plan. If the goal image is too far in the future, the planning problem is too hard, but if the goal image is too similar to the current observation then there isn’t much signal for how to progress.
* In stage 1, learning the latent planner requires a multi-view camera setup to collect human play data. Is this assumption carried through to stage 2 and testing? If so, it seems like a strong assumption to have a calibrated multi-view camera setup all the time at deployment / testing time. How does this method perform with only a single-view camera setup?
* In the “transformer-based plan-guided imitation” section it seems like the 3D end-effector representation is not used for the robot (only in phase 1 for the human). Why get rid of this intermediate representation? This would make the cross-embodiment problem easier (e.g., as done in Bahl et al, “Human-to-Robot Imitation in the Wild” RSS 2022).
* It seems like many of the core ideas from the prior work [34] “Third-person visual imitation learning via decoupled hierarchical controller” and the proposed work are quite similar (e.g., learning from human video demos a hierarchical model that decouples goal prediction from low-level control). One of the core differences is the human play data, and then several design differences. Why was this method not benchmarked against and how do you think it would compare?


**Robotics Focus:**

Sufficient demonstration on hardware

**Summary Of Paper:**

This paper combines human play data—video sequences of people interacting with the environment in an unstructured way–and teleoperation data to teach a robot long-horizon skills. Using human play data, one can first learn a latent planner that takes as input raw video of a human (or robot) and infers a representation that enables high-quality reconstruction of the agent’s end-effector trajectory. The robot’s low-level motor control policies are trained from teleoperation data which perform the same task, but are conditioned on the output of the pre-trained latent planner. These two modules are then combined during testing where a long-horizon video prompt is given to the robot and it generates latent plans and low-level controls to accomplish the task. The paper showcases favorable quantitative and qualitative performance on many real-world robotics tasks and in hardware on a Franka robot.

**Summary Of Recommendation:**

Although I have several questions I'd like to have addressed during the rebuttal about the underlying assumptions of this method, I believe this is a strong and relevant contribution to CoRL. Thus, my recommendation is accept.

---

### Official Review · Reviewer_mxMa · 2023-07-16

**Confidence:** 4
**Originality:** Excellent
**Technical Quality:** Very Good
**Clarity Of Presentation:** Good
**Impact:** 4

**Recommendation:**

Strong Accept: I recommend accepting the paper and will argue for my recommendation even if other reviewers hold a different opinion.

**Review:**

**Summary**:

This work introduces MimicPlay, a hierarchical learning from play data framework that decouples high-level latent plan learning from low-level control learning. The motivation is that in the standard robot play setting, both high level plans as well as low-level control must be learned from the same data source (teleoperated robot play data) despite these two problems having very different problem requirements. Thus, MimicPlay uses human play data in the same domain to learn high level latent plans, which are then used by a low-level policy that learns from a much smaller amount of teleoperated robot data (<30 demos) compared to prior works. The high level planner learn to predict future human hand 3D poses given the initial state and a goal state (10-30s in future) with a GMM to handle the inherent multimodality of play data, coupled with a KL divergence loss to enforce learned visual encoders to minimize the distribution shift between robot and human domain data. The low-level policy is a standard latent goal conditioned multitask imitation learning policy. There are quite extensive experiments on 14 tasks across 6 different environments, including comparisons with baselines such as GC-BC, LMP, C-BeT, and R3M; in these experiments, MimicPlay is the most performant, especially in the lower data regime. In addition, an ablation study shows that the multimodal GMM is crucial for performance, as well as the human data.

**Strengths**:
- This work is strongly motivated, since the expense of collecting real data is non-trivial and any work that reduces the data requirement (such as by offloading high-level plan learning to much cheaper human data) will be impactful if successful.
- The empirical results are extensive, comparing to baseline across other play methods (ie. LMP and C-BeT) as well as representation learning methods (R3M).
- The technical soundness is strong, with the ablations showing the importance of the GMM as well as the human data, both of which are core claims of the method.
- The method is elegant and performant, with the cross-morphology results with prompting from either human or robot videos very compelling.

**Weaknesses**:
- The hand orientations are 3D only without orientation information. Since the 3D pose prediction (opposed to 2D from prior works) is noted as a minor contribution, it would be helpful to #1 study the impact of expanding 2D to 3D plans and #2 clarify why orientation is not included in the plans.
- The KL term for bridging the robot visual distribution to the human visual distribution may not scale when both of these distributions become more complex. For example, robot data may become more challenging with in a more complex setting (ie. mobile manipulation) or with more realistic hardware (ie. un-calibrated camera) or with more difficult tasks (ie. home setting). And in parallel, human data may become more challenging with more diverse environments (ie. Ego4D / EpicKitchens) or realistic hardware (ie. no 3D ground-truth pose if there are no calibrated RGBD cameras). Therefore, I am worried that the KL term alone may not scale well to more realistic settings.
- Related to the previous point, not only will the visual distribution shift cause challenges with scale, also the plans themselves may only be useful if the human play data is in the exact same domain as the robot demos.
- Some implementation details were not clear in the main work. For example, data requirements for the human play as well as robot demos (how many?) were not clear. Or, it's unclear what is meant for "baselines get 10x more demos" -- is this 10x robot demos that are used for both plan learning and low-level control learning? For the robot data, did they receive demos or play data (my understanding is MimicPlay does not use any robot play data but only demos; on the other hand, the baseline play methods do not use any demos but only robot play data)

**Quality Of The Limitations Section:**

Limitations are addressed clearly

**Questions For Rebuttal:**

My main questions would be to clarify my points raised in the weakness section above.

**Robotics Focus:**

Sufficient demonstration on hardware

**Summary Of Paper:**

(see below for summary)

**Summary Of Recommendation:**

This work is well-motivated and performant, with strong potential impact in an order of magnitude reduction with on-robot data required. There are extensive empirical investigations on multiple domains with baselines from different domains. The presentation is fairly good, but minor improvements and clarifications can be made. I also have a few minor points about the future scalability of the method, but it does not detract much from my holistic rating. Overall, I believe the work is impactful and significant, and makes a strong contribution to hierarchical robot learning under real world data considerations.

[Post-Rebuttal Update]
The reviewers have addressed my questions and I maintain my positive rating of Strong Accept. I believe this is a valuable contribution and strongly argue for acceptance.

---

### Official Review · Reviewer_17ab · 2023-07-18

**Confidence:** 4
**Originality:** Good
**Technical Quality:** Good
**Clarity Of Presentation:** Fair
**Impact:** 3

**Recommendation:**

Weak Accept: I recommend accepting the paper, but will not argue for my recommendation if the majority of other reviewers have a different opinion.

**Review:**

Strengths:
- The method does grow on an interesting pair of times in the community about how to take better advantage of data that is much cheaper to collect, in particular data of humans interacting with their own hands inside of the workspace that the robot arm will end up operating in.
- The method also proposes using a GMM model to help capture the multimodality of human demonstrations better.
- This appears to result in some strong performance for the overall method.

Weaknesses:
- There are a number of papers that use a similar paradigm of trying to train robotic manipulation models from play data. The paper would benefit from situating itself amongst this literature better.
- There are many components of the method section that are not explained clearly enough to understand how the method is designed.
- More information on how the methods are compared and how performance is defined will shed more light on understanding the performance differences between these method environments.


-------
I have updated my score after three discussions.

**Quality Of The Limitations Section:**

Additional details required

**Questions For Rebuttal:**

Comments:
- It's not easy to understand the details about how the cross-modal representation is learned for a model that works both for human play data and for robotics data. There only seem to be a few sentences on this topic online 162. this seems to be an important feature or hurdle that the modal needs to overcome in order to be successful. Without these details, it's difficult to understand how the model should work properly.
- In addition, the paper is missing information about the one-shot video prediction method that is used to help generate trajectories. this is described online 184; there should at least be citations towards the models that are used to do this contextual video generation. How is this video prompting being performed? How are the prompts being designed such that it will result in the video generation model producing a better plan?
- One question that comes to mind while reading the paper is whether or not there is enough data to train the transformer model? Transformers often need much larger amounts of data to be very successful so it's unclear if the use of the transformer in this paper is truly helpful compared to the amount of data collected for the experiments.
- The concept that training a latent model helps increase the success of planning over longer Horizons is not a novel contribution. this has been studied in many prior papers and has been shown to be increasingly important, especially for longer Horizon tasks.
- The results shown in the tables are missing important information and should include more significant figures. To all readers to better process the results of the experiments, more details are the quantities should be reported as well as confidences/variances over the computed quantities.
- For showing the importance of the GMM method starting in line 246, the authors note that the method is very important for providing better trajectory generation. The authors need to provide more information on how they determine which trajectory generation is better. what type of metric is used to define this type of better trajectory generation? This should be in the main paper.


**Robotics Focus:**

Highly relevant to robotics but no hardware experiments

**Summary Of Paper:**

This paper proposes a system for developing a stronger method for long Horizon manipulation planning. This has been an area of difficulty for the community for many years, especially when applying learning-based methods that struggle to be able to deal with the exploration challenges of solving long Horizon tasks. The method proposes a two-level process where in the first level, a large amount of human interaction data is collected, and a latent model is learned to be able to generate trajectories from this human experience. Then from this human experience, data and additional model is trained to be able to prompt the model to generate good plans to solve individual tasks. The method does show some promising performance on real hardware, but there are details missing from the paper that makes it difficult to understand the method.

**Summary Of Recommendation:**

The paper proposes a potentially solid method to enable the community to train robotic controllers to perform better at long-horizon planning problems. However, the writing lacks details, making it difficult to fully understand the proposed method. If these details are addressed it will improve the paper.

---

### Decision · Program_Chairs · 2023-08-30

**Decision:**

Accept (Oral)

**Comment:**

This paper presents a method which uses a combination of human play data and robot teleoperation data, in order to train manipulation policies that address a variety of tasks, including some of reasonably long horizon. This paper addresses several important themes in robot learning, including: (i) it benefits from using third-person data of humans, (ii) it benefits from using data which is not necessarily task-specific ("play" data), (iii) it shows relevance to "long horizon" manipulation tasks, (iv) the human demos can be thought of as a type of multimodal prompting for addressing manipulation tasks.

In addition to the relevance of its general topics, the authors work was generally well received by the mix of reviewers.  Multiple reviewers commented on the strength of the motivation and the value of the empirical results.  3 reviewers started with Strong Accepts, and after a robust rebuttal discussion with the 4th reviewer, they increased to Weak Accept.  One of the key differentiators that came up in the rebuttal with the 4th reviewer, is that this work uses not just (a) "human" data i.e. of task-specific human demonstrations, or (b) "play" data i.e. of robots collecting non-task-specific data, but it specifically uses (a+b) "human play" data, in which humans are collecting non-task-specific data.

There are many other details of this paper that provide valuable technical ideas and analysis, including the details of the high-level/low-level policy interaction, the 3D human hand tracking, and reasonably comprehensive ablations.

Despite overall complementary reviews, the authors still have several things to do to best incorporate all feedback from reviewers, including adding more seeded evaluations (which was mentioned by multiple reviewers) and describing the additional missing details discussed.  Also the authors mention in the rebuttal they plan to open-source their code, and I would highly recommend to do so it they would like to maximize the impact of their work.